# Molecular epidemiology of *Leptospira* spp. among wild mammals and a dog in Amami Oshima Island, Japan

So Shinya₀[1]*, Yukinori Muraoka[2], Daigo Negishi[1], Nobuo Koizumi₀[3]*

1 Yuinoshima Animal Clinic, Amami-City, Kagoshima, Japan, 2 Shintoshin Animal Clinic, Saitama-City, Saitama, Japan, 3 Department of Bacteriology I, National Institute of Infectious Diseases, Shinjuku-ku, Tokyo, Japan

* abzeoz@gmail.com (SS); nkoizumi@niid.go.jp (NK)

## Abstract

Leptospirosis is a worldwide zoonosis caused by the pathogenic *Leptospira* spp. Canine and human leptospirosis sometimes occur on Amami Oshima Island, located in the Nansei Archipelago, southwestern Japan; however, information on the causative *Leptospira* spp. on this island is quite limited. This study aimed to investigate the molecular and serological characteristics of *Leptospira* spp. isolated from wild animals and a dog in Amami Oshima Island. We obtained seven *Leptospira* strains by culturing kidney tissues of wild animals, such as black rats (2), wild boars (3), and rabbit (1) as well as blood from a symptomatic dog. Using *flaB* sequencing and microscopic agglutination test with antisera for 18 serovars, the isolates were identified as *Leptospira borgpetersenii* serogroups Javanica (black rat), *L. interrogans* serogroup Australis (black rat and dog), and *L. interrogans* serogroup Hebdomadis (wild boar and rabbit). The sequence type (ST) of *L. borgpetersenii* serogroup Javanica was determined to be ST143 via multilocus sequence typing (MLST) using seven housekeeping genes. For *L. interrogans*, MLST and multiple-locus variable-tandem repeat analysis (MLVA) revealed identical ST and MLVA types in rat and canine isolates, whereas two STs and MLVA types were identified in wild boar isolates. The STs and MLVA types of rabbit and one of the wild boars were identical. Bacterial culture and *flaB*-nested polymerase chain reaction demonstrated a high rate of *Leptospira* infection in wild boars (58.3%, 7/12), whereas *Leptospira* spp. were detected in 4.8% of black rats (2/42). This study revealed diverse *Leptospira* genotype and serotype maintenance in wild mammals on Amami Oshima Island. MLST and MLVA indicated that black rats were a source of canine infection. Wild boars carry *L. interrogans* and are considered an important maintenance host because antibodies against serogroup Hebdomadis were detected in human and canine leptospirosis patients on this island.

## Introduction

Leptospirosis is a worldwide zoonosis caused by infection with pathogenic spirochetes belonging to the genus *Leptospira* [1,2]. Pathogenic *Leptospira* spp. colonize the proximal renal

---

**Data Availability Statement:** All relevant data are within the manuscript and its Supporting Information files.

**Funding:** Nobuo Koizumi Agency: the Japan Agency for Medical Research and Development

(AMED) Fund no.: JP19fk0108049 Nobuo Koizumi Agency: the Japan Agency for Medical Research and Development (AMED) Fund no.: JP20fk0108139 This work was supported by the Research Program on Emerging and Re-emerging Infectious Diseases (JP19fk0108049 and JP20fk0108139 from the Japan Agency for Medical Research and Development (AMED) (NK).

**Competing interests:** The authors have declared that no competing interests exist.

tubules of animal reservoirs and are excreted in the urine. Dogs and other mammals, including humans, are infected with bacteria through direct contact with the urine of reservoir animals or with the environment, such as water and soil contaminated by their urine [2]. Leptospirosis is one of the most important canine infectious diseases, and dogs exhibit acute or subacute hepatic and renal failure [3]. Humans contract the disease, especially when they are engaged in agricultural work or leisure in freshwaters, such as rivers or lakes [4].

*Leptospira* spp. are classified into four subclades, P1, P2, S1, and S2, based on genome sequences, of which species belonging to P1 and P2 can cause leptospirosis [5,6]. *Leptospira* spp. are divided into serovars based on their antigenic properties, and antigenically related serovars are grouped into serogroups [4]. Generally, each serovar is associated with a particular maintenance host. For example, serovar Icterohaemorrhagiae is maintained by *Rattus* species, serovar Hardjo by cattle, and serovar Canicola by dogs [1,4,7]. In contrast to the above specific serovar-animal association, multiple serovars/serogroups in a single animal species have been observed in small Indian mongoose or rats [1,8]. Recently, molecular typing methods such as multilocus sequencing typing (MLST) and multiple-locus variable-number tandem repeat analysis (MLVA) have been demonstrated to be useful tools for identifying *Leptospira* serovars/serogroups or discriminating strains within the same serovar [9–15]. These molecular typing methods indicated an association between virulence and specific *Leptospira* genotypes [12–14,16,17].

Amami Oshima Island is located in the Nansei Archipelago and in a subtropical zone where people frequently have contact with the natural environment or wild mammals because leisure in rivers and wild mammal hunting involving hounds are popular. Although two cases of human leptospirosis were reported in 2016 [18,19], and several dogs were clinically suspected to be infected each year, *Leptospira* spp. have never been isolated from human patients and wild or domestic animals. Furthermore, prevalent *Leptospira* genotypes and serotypes remain unknown. In this study, we isolated and detected *Leptospira* spp. from wild and feral animals, as well as in symptomatic dogs. We characterized *Leptospira* isolates using the microscopic agglutination test (MAT) with antisera for 18 serovars and MLST or MLVA.

## Materials and methods

### Sampling sites and animals

The present study was conducted on Amami Oshima Island, located in the Nansei Archipelago, southwestern Japan. The island is located at a latitude of 28.19˚ N and a longitude of 129.22˚ E. It has a population of approximately 58 thousand people. Sampling sites of wild and feral animals, as well as places of residence of symptomatic dogs, are shown in Fig 1.

A total of 42 black rats (*Rattus rattus*) were captured as a bycatch under the extermination program of invasive feral cats conducted by the Ministry of the Environment, and this investigation was approved by the Ministry. Twelve Ryukyu wild boars (*Sus scrofa riukiuanus*) and two goats (*Capra aegagrus hircus*) were captured by local licensed hunters, and kidney tissues were provided by them. An Amami rabbit (*Pentalagus furnessi*) killed by a traffic accident and collected under the rescue program of injured wild animals conducted by the Ministry of the Environment was used in this study. The cultivation of *Leptospira* spp. from dead Amami rabbits was also approved by the Ministry. Four dogs with clinically suspected leptospirosis were included in this study. Blood was collected from the dogs that exhibited (i) at least two of the four symptoms: fever, vomiting, hyperemia, and hemorrhage of the mucous membranes and jaundice; and/or (ii) acute renal involvement (abnormal values of creatinine and/or blood urea nitrogen) of unknown origin; and/or (iii) acute hepatic involvement (abnormal values of alanine aminotransferase, aspartate aminotransferase, and/or alkaline phosphatase) of unknown

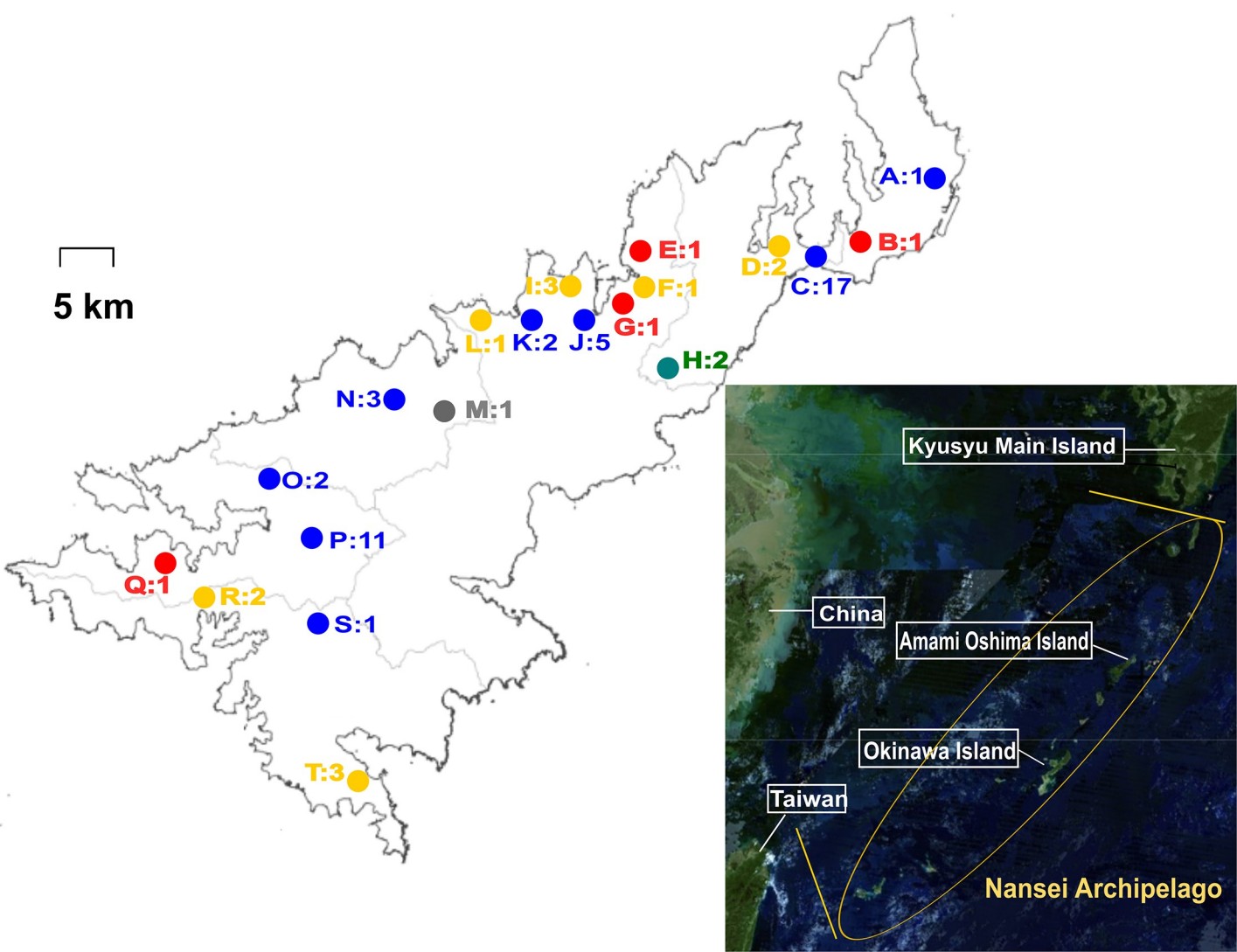

**Fig 1. Sampling sites of wild and feral animals as well as places of residence of symptomatic dogs in this study.** Circles indicate the sampling sites of wild and feral animals (A, C, D, F, H–P, and R–T) and places of residence of symptomatic dogs (B, E, G, and Q). Each color represents animal species: blue, black rats; yellow, Ryukyu wild boars; red, dogs; white, goats; gray, Amami rabbit. The numbers indicate the number of samples collected at each site.

origin. The consent for collection of blood for *Leptospira* cultivation was orally obtained from the owners. The samples were taken from November 2019 to September 2020.

## Cultivation of *Leptospira* spp. from animals

Five square millimeters of the cortex of the kidney tissue of black rats, Ryukyu wild boars, goats, and Amami rabbit were ground using a sterile 2.5 mL disposable syringe and inoculated into 4 mL of liquid modified Korthof's medium with 10% rabbit serum [2] and settled at ambient temperature (20–30˚C) overnight. Five hundred microliters of the culture supernatant were inoculated into 4 mL of Korthof's medium the next day and cultured at 30˚C for 3 months. Rats were euthanized by isoflurane inhalation and processed according to the American Veterinary Medical Association guidelines. The isolation of *Leptospira* spp. from black rats was approved by the Ministry of the Environment. One hundred microliters of blood from a

symptomatic dog were inoculated into 4 mL of Korthof's medium and cultured as described above. Growth of *Leptospira* spp. was observed under a dark-field microscope every week in the first month and once every two weeks. The culture was judged as negative when the growth of *Leptospira* spp. was not observed after 3 months.

## Species classification of *Leptospira* isolates

Genomic DNA was extracted from 0.5 mL cultures of the isolates using the DNeasy Blood and Tissue Kit (QIAGEN, Hilden, Germany). The partial *flaB* gene was amplified with *Takara EX Taq* Hot Start version (Takara Bio, Japan) using the primer set L-*flaB*-F1 and L-*flaB*-R1 as previously described [15]. The polymerase chain reaction (PCR) products were purified with ExoSAP-IT Express PCR Cleanup Reagents (Applied Biosystems, Foster City, CA, USA) according to the manufacturer's instructions. Direct cycle sequencing of the purified PCR products was performed using the BigDye Terminator v3.1 Cycle Sequencing Kit (Applied Biosystems, Foster City, CA, USA) with the primers, and the resulting sequences were compared to those of reference strains through BLAST search (https://blast.ncbi.nlm.nih.gov/Blast.cgi). The *flaB* sequences were deposited in a public database (DDBJ accession numbers LC596935–LC596941).

## Serogroup identification of *Leptospira* isolates

Serogroups of the isolates were identified via MAT using a panel of anti-*Leptospira* rabbit sera for 18 serovars [12]. Twenty-five microliters of 4- to 7-day-cultures of the isolates were incubated with the same volume of 200-fold antisera diluted with PBS or PBS alone for 2.5 h at 30°C, and the reaction was interpreted as positive when the proportion of free, unagglutinated leptospires was <50% compared to the control suspension.

## MLST

MLST was performed for the *Leptospira* strains isolated in this study, for those isolated in Kyushu Main Island, and those isolated in Okinawa Prefecture (Table 2) using seven housekeeping genes, *glmU*, *pntA*, *sucA*, *tpiA*, *pfkB*, *mreA*, and *caiB*, as previously described [20]. Sequence type (ST) was assigned using the PubMLST database [21]. Information on the isolates analyzed in this study and new STs (ST328, ST329, and ST331) obtained in this study have been deposited in the database. Phylogenetic relationships of the concatenated sequences (5'-*glmU-pntA-sucA-tpiA-pfkB-mreA-caiB*-3') determined in this study and deposited in the MLST database (PubMLST, https://pubmlst.org/organisms/leptospira-spp) were determined by reconstructing a phylogenetic tree by the maximum likelihood method using IQ-TREE [22] with 1,000 ultrafast bootstrap replicates.

## MLVA

MLVA was performed for *L. interrogans* strains isolated in this study and in Okinawa Prefecture using 11 loci, variable number tandem repeat (VNTR) 4, 7, 10 [23], 19 [9], 23 [11], 31 [9], 27, 29, 30, 36, and 50 [10], as previously described [13]. The amplified products were electrophoresed on 2% agarose gel, and their sizes were estimated by comparison with a 100 bp ladder DNA marker or were determined via DNA sequencing as mentioned above. The genome sequences containing VNTR determined in this study have been deposited in a public database (DDBJ accession numbers LC619028–LC619057).

### Leptospiral DNA detection from kidney tissue of wild and feral animals

Genomic DNA was extracted from the kidney tissues of 41 rats, 12 wild boars, two goats, and a rabbit, using the DNeasy Blood and Tissue Kit (Qiagen). The leptospiral *flaB* gene was detected via nested PCR as previously described [12], followed by DNA sequencing with the second PCR primers of the nested PCR as described above.

## Results

### Species and serogroups of *Leptospira* isolated from wild mammals and a dog in Amami Oshima Island

*Leptospira* spp. were isolated from three of the 12 Ryukyu wild boars (25%), two of the 42 black rats (4.8%), one Amami rabbit (100%), and one of the four dogs with clinically suspected leptospirosis (25%, S1 Table). No isolates were obtained from goats. The isolates were identified as *L. interrogans* serogroup Hebdomadis (four strains from three wild boars and a rabbit, sampling sites D, I, and M in Fig 1), *L. interrogans* serogroup Australis (two from a black rat and a dog, B and C), and *L. borgpetersenii* serogroup Javanica (one from a black rat, K) (Table 1). The *flaB* sequences were identical for each serogroup.

### Molecular typing of *Leptospira* isolates from wild mammals and a dog in Amami Oshima Island

MLST using seven housekeeping genes revealed that *L. borgpetersenii* AMM-001 (rat isolate) belonged to ST143. The rat and dog isolates of *L. interrogans* serogroup Australis (AMM-012 and D-KS19-7K) belonged to ST105, which was detected in Okinawa Prefecture and Taiwan, whereas the *L. interrogans* serogroup Australis strain isolated from Kyushu Main Island belonged to the phylogenetically distant lineage, ST120 (Table 2 and S1 Fig). Two STs were identified in *L. interrogans* serogroup Hebdomadis isolates (Table 2). Two of the three Ryukyu wild boar isolates (AMM-042 and AMM-043) showed the novel ST, ST328, which was phylogenetically related to other STs of serogroup Hebdomadis strains in Japan (S1 Fig). The other wild boar isolate (AMM-047) and the Amami rabbit isolate (AMM-057) belonged to ST140, which has been identified in *L. interrogans* serogroup Hebdomadis strains on Kyushu Main Island (Table 2).

MLVA was further employed for *L. interrogans* isolates because MLVA using 11 loci has higher discriminatory power and was more concordant with serotyping than MLST using seven housekeeping genes for *L. interrogans* [13,14]. The rat and dog isolates of *L. interrogans* serogroup Australis (AMM-012 and D-KS19-7K) showed the same MLVA profile (MLVA type) (Table 2). The profile was almost identical to that of the isolates from Okinawa Prefecture and Taiwan (identity: 10/11, Table 2), whereas it was different from those from Kyushu Main Island (identity: ~1/11) (Table 2) [13,14]. As with MLST, two MLVA types were identified in *L. interrogans* serogroup Hebdomadis isolates: the wild boars and rabbit isolates (AMM-047 and AMM-057) showed identical MLVA profiles, which were identical to those of the isolates from Kyushu Main Island, whereas they were different from those from Okinawa Prefecture (identity: ~3/11, Table 2) [13,14]. The MLVA types of AMM-042 and AMM-043 were different from those of the isolates from Kyushu Main Island and Okinawa Prefecture (identity: ~4/11, Table 2) [13,14].

### Leptospiral DNA detection from kidney tissues of wild and feral animals

Fifty-six kidney samples (41 black rats, 12 wild boars, two goats, and a rabbit) were subjected to nested PCR, and leptospiral *flaB* was detected in two rat samples (sampling sites C and K in

Fig 1), four wild boar samples (T and R), and a rabbit sample (M). Although the PCR-positive rat and rabbit samples were culture-positive, four PCR-positive wild boar samples were all culture-negative, and three culture-positive samples were all PCR-negative, suggesting degradation of DNA during storage. In combination with culture and PCR results, *L. interrogans* was detected in seven of the 12 Ryukyu wild boars (58.3%). Three *flaB* sequences detected in the wild boar samples were identical to those of the wild boar isolates, whereas the other sequence was identical to that of the isolates except one doublet peak, which was probably due to mixed infection.

## Discussion

### Reservoir animals of *Leptospira* spp. in Amami Oshima Island

In this study, *L. borgpetersenii* and *L. interrogans* were isolated from wild mammals such as black rat, Ryukyu wild boar, and Amami rabbit. The rat and dog isolates of *L. interrogans* serogroup Australis (AMM-012 and D-KS19-7K) showed the same MLVA profiles. Furthermore, the sampling sites of the rat and dog were almost the same (sites B and C in Fig 1), strongly suggesting that the dog acquired *L. interrogans* from black rats or the environment contaminated by their urine. Therefore, similar to previous studies conducted in other areas [24,25], black rats may be an important reservoir of *L. interrogans* for dogs on Amami Oshima Island. Although infection of humans with *L. interrogans* serogroup Australis has never been reported on this island, the identical ST and the similarity of MLVA profiles between the rat/dog isolates and the human isolate in Okinawa Prefecture (Table 2) indicated that this strain can cause leptospirosis in humans.

This study demonstrated the high frequency of *Leptospira* infection in wild boars on this island: 58.3% of the wild boars examined were culture- or PCR-positive in different areas of Amami Oshima Island. This positive rate seems higher than that in other areas of Japan: 15.2% PCR-positive in wild boar kidney samples [26]. Although *L. borgpetersenii* serovar Tarassovi and *L. interrogans* serovars Australis, Bratislava, Icterohaemorrhagiae, and Pomona have been isolated from European wild boar species in Croatia and Italy [27,28], *L. interrogans* serogroup Hebdomadis was identified in wild boars for the first time. The *flaB* sequences detected from kidney samples using nested PCR were identical to those of the *L. interrogans* serogroup Hebdomadis isolates. Although no *Leptospira* isolates were obtained, anti-Hebdomadis antibodies were detected in humans and dogs with leptospirosis on this island (S2 Table) [18]. These results indicate that wild boars are maintenance hosts of *L. interrogans* serogroup Hebdomadis and a source of human and dog infections. In this study, the four PCR-positive wild boar samples were all culture-negative, and three culture-positive samples were all PCR-negative. The difference between PCR and culture results in this study was probably due to the lower sensitivity of culture than PCR [29], as well as the possible degradation of DNA during storage.

*L. interrogans* serogroup Hebdomadis was isolated from an Amami rabbit for the first time, which is registered as an endangered by the IUCN Red List of Threatened Species and inhabits only Amami Islands [30]. In Europe, wild rabbits possess high levels of antibodies against *Leptospira* spp., and Grippotyphosa is the most prevalent serogroup [31,32]. Mild to moderate

**Table 1. Species and serogroups of *Leptospira* isolates obtained in this study.**

| Species | Serogroup | Amami rabbit (n = 1) | Black Rat (n = 42) | Goat (n = 2) | Dog (n = 4) | Ryukyu wild boars (n = 12) |
|---|---|---|---|---|---|---|
| *L. borgpetersenii* | Javanica | | 1 | | | |
| *L. interrogans* | Australis | | 1 | | 1 | |
| | Hebdomadis | 1 | | | | 3 |

**Table 2. STs and MLVA profiles of *L. interrogans* isolates analyzed in this study.**

| Strain | Serogroup | Animal | Capture/Residence site[a] | MLSTST | VNTR | | | | | | | | | | |
|---|---|---|---|---|---|---|---|---|---|---|---|---|---|---|---|
| | | | | | 4 | 7 | 10 | 19 | 23 | 27 | 29 | 30 | 31 | 36 | 50 |
| AMM-012 | Australis | Black rat | C | 105 | 7 | 9 | 13 | 14 | 6 | 10 | 7 | 8 | 1 | 11 | 8 |
| AMM-042 | Hebdomadis | Wild boar | D | 328 | 2 | 9 | 10 | 10 | 4 | 13 | 6 | 7 | 1 | 9 | 4 |
| AMM-043 | Hebdomadis | Wild boar | D | 328 | 2 | 9 | 10 | 10 | 4 | 13 | 6 | 7 | 1 | 9 | 4 |
| AMM-047 | Hebdomadis | Wild boar | I | 140 | 3 | 13 | 9 | 11 | 2 | 3 | 4 | 8 | 3 | 4 | 4 |
| AMM-057 | Hebdomadis | Rabbit | M | 140 | 3 | 13 | 9 | 11 | 2 | 3 | 4 | 8 | 3 | 4 | 4 |
| D-KS19-7K | Australis | Dog | B | 105 | 7 | 9 | 13 | 14 | 6 | 10 | 7 | 8 | 1 | 11 | 8 |
| OP098067 | Hebdomadis | Human | _[h] | 118 | 1 | 4 | 5 | 8 | 0 | 3 | 9 | 5 | 3 | 11 | 4 |
| OP82 | Hebdomadis | Human | _[h] | 118 | 1 | 4 | 5 | 7 | 0 | 3 | 9 | 5 | 5 | 11 | 4 |
| OW09-1K[b] | Hebdomadis | Dog | _[h] | 295 | 1 | 13 | 7 | 10 | 3 | 12 | 7 | 8 | 1 | 7 | 4 |
| OR108-1[b] | Hebdomadis | Mouse[d] | _[h] | 295 | 1 | 13 | 7 | 10 | 3 | 12 | 7 | 8 | 1 | 7 | 4 |
| OP98 | Hebdomadis | Human | _[h] | 295 | 1 | 13 | 7 | 10 | 3 | 12 | 7 | 8 | 1 | 7 | 4 |
| OHJ2008-92U | Hebdomadis | Mongoose[e] | _[h] | 295 | 1 | 13 | 7 | 10 | 3 | 12 | 7 | 8 | 1 | 7 | 4 |
| OP118096 | Hebdomadis | Human | _[h] | 329 | 2 | 13 | 7 | 10 | 3 | 12 | 7 | 8 | 4 | 7 | 4 |
| OP83 | Hebdomadis | Human | _[h] | 329 | 2 | 13 | 7 | 10 | 3 | 12 | 7 | 8 | 5 | 7 | 4 |
| OHJ2008-135U | Hebdomadis | Mongoose[e] | _[h] | 329 | 2 | 13 | 7 | 10 | 3 | 12 | 7 | 8 | 6 | 6 | 4 |
| OHJ2010-G6U | Hebdomadis | Mongoose[e] | _[h] | 329 | 2 | 13 | 7 | 10 | 3 | 12 | 7 | 8 | 7 | 7 | 4 |
| OP43 | Hebdomadis | Human | _[h] | 35 | 2 | 12 | 5 | 6 | 3 | 12 | 6 | 8 | 4 | 6 | 4 |
| OP088041 | Australis | Human | _[h] | 105 | 7 | 9 | 13 | 6 | 6 | 10 | 7 | 8 | 1 | 11 | 8 |
| R725[b] | Australis | Rat[f] | _[i] | 105 | 7 | 9 | 13 | 12 | 6 | 10 | 7 | 8 | 1 | 11 | 8 |
| D-SA11-7E[b] | Hebdomadis | Dog | _[j] | 36 | 3 | 13 | 16 | 10 | 3 | 9 | 6 | 2 | 2 | 5 | 4 |
| D-MZ07-13K[b] | Hebdomadis | Dog | _[j] | 37 | 1 | 5 | 5 | 8 | 0 | 11 | 9 | 11 | 1 | 11 | 4 |
| D-MZ10-8E[b] | Hebdomadis | Dog | _[j] | 118 | 1 | 4 | 5 | 9 | 3 | 3 | 9 | 11 | 1 | 11 | 4 |
| D-MZ10-17E[b] | Hebdomadis | Dog | _[j] | 119 | 2 | 13 | 7 | 10 | 3 | 12 | 7 | 8 | 1 | 7 | 4 |
| D-KM11-3E[b] | Hebdomadis | Dog | _[j] | 138 | 2 | 13 | 5 | 10 | 3 | 12 | 7 | 8 | 6 | 7 | 4 |
| D-MZ08-2E[b] | Hebdomadis | Dog | _[j] | 140 | 3 | 13 | 9 | 11 | 2 | 3 | 4 | 8 | 3 | 4 | 4 |
| AS-KS09-29[b] | Hebdomadis | Mouse[g] | _[j] | 140 | 3 | 13 | 9 | 11 | 2 | 3 | 4 | 8 | 3 | 4 | 4 |
| D-KS10-5E[b] | Hebdomadis | Dog | _[j] | 331 | 2 | 13 | 5 | 10 | 3 | 12 | 7 | 8 | 1 | 7 | 4 |
| D-MZ07-5K[b] | Australis | Dog | _[j] | 37 | 3 | 7 | 3 | 10 | 0 | 6 | 14 | 3 | 6 | 0 | 4 |
| D-MZ07-15E[b] | Australis | Dog | _[j] | 37 | 4 | 6 | 3 | 11 | 0 | 7 | 15 | 3 | 2 | 0 | 4 |
| D-FO11-11K[b,c] | Australis | Dog | _[j] | 120 | 2 | 7 | 3 | 10 | 0 | 6 | 15 | 3 | 4 | 0 | 4 |

[a] Capture/residence sites are depicted in Fig 1.

[b] MLVA profiles of these strains were determined in previous studies [13,14].

[c] The STs of these strains were determined in a previous study [20].

[d] *Mus caroli*.

[e] *Herpestes auropunctatus*.

[f] *Rattus losea*.

[g] *Apodemus speciosus*.

[h] captured in Okinawa Prefecture.

[i] captured in Taiwan.

[j] captured in Kyushu Main Island.

interstitial nephritis and necrosis of renal tubules were observed in wild rabbits in Nigeria, from which *L. interrogans* serovars Bratislava, Canicola, and Icterohaemorrhagiae were isolated [33]. The MLVA types of wild boar and rabbit isolates were identical in this study (Table 2). More research is necessary to determine whether Amami rabbits are an incidental

or maintenance host of *L. interrogans* serogroup Hebdomadis, as well as conservation strategies for this endangered species.

*L. borgpetersenii* serogroup Javanica ST143 isolated from black rats in this study is distributed widely in rats in the Nansei Archipelago and in Asian countries [23,34]. Although there are no reports on infection of humans or dogs with this strain on this island, ST143 was isolated from leptospirosis patients in Okinawa Prefecture and the Philippines [19,23]. Anti-Javanica antibodies have been detected in canine sera using MAT in Malaysia and Thailand [34–36].

## Genetic comparison of *Leptospira* isolates in Amami Oshima Island and neighboring areas

This study revealed that the rat isolates were genetically more closely related to strains on Okinawa Island than those on Kyushu Main Island, regardless of *Leptospira* species or serogroup. In Japan, *L. borgpetersenii* Javanica ST143 has been detected only in the Nansei Archipelago, whereas it is widely distributed in Asian countries [14,23]. The ST and MLVA profiles of *L. interrogans* serogroup Australis were the same and almost the same as those of the Okinawa and Taiwan isolates, but different from those of isolates from Kyushu Main Island (Table 2) [13,14]. Amami Oshima Island was connected to the southwestern part of China, Taiwan, and Okinawa, but not to other parts of Japan, including Kyusyu Main Island. Amami Oshima Island had then separated from other islands 1.2 million years ago, and organisms have evolved separately from Kyushu Main Island and other Japanese islands [37]. Consequently, many closely related endemic species from Amami Oshima Island and Okinawa Island inhabit this region [38,39]. Host animals and the environmental influence on them are important factors in *Leptospira* diversification [14,40,41], which is a possible reason why serogroups Australis and Javanica isolates were closely genetically related to those in Okinawa Islands. Indeed, geographical structuring of genetic diversity has been reported for *L. borgpetersenii* serogroup Javanica in southern Japan, the Philippines, and Taiwan [34].

Ryukyu wild boars inhabit the Nansei Archipelago and differ from wild boars in other areas of Japan [42]. A serosurvey found that Hebdomadis is the most prevalent serogroup among wild boars on Okinawa Island [43]. In this study, however, the MLVA type/ST140 of serogroup Hebdomadis isolated from wild boars (and an Amami rabbit) were identified from those from Kyusyu Main Island and not Okinawa Islands (Table 2) [13,14]. The settlement of large Japanese field mice from which this MLVA type was isolated on Kyusyu Main Island (Table 2) was not confirmed on this island. Dogs were brought to this island from Kyusyu Main Island, where this MLVA type was highly isolated from dogs with leptospirosis [12,13]. Although chronic carriage of this MLVA type of *L. interrogans* serogroup Hebdomadis in asymptomatic dogs has not been proven, it might be introduced with dogs from Kyushu Main Island and spread among wild boars. On the other hand, the same ST, ST118, was identified in human and dog patients both in Kyushu Main Island and Okinawa Prefecture (Table 2), suggesting that there is an unidentified maintenance animal(s) that carries this ST in these areas. This may also be true for the MLVA type/ST140 of the serogroup Hebdomadis strain. In addition, novel ST/MLVA profiles were identified in wild boar isolates of the serogroup Hebdomadis. Further research is needed to clarify the phylogenetic relationship of *L. interrogans* serogroup Hebdomadis strains in Japan.

## Prevention of leptospirosis in Amami Oshima Island

This study revealed that various wild mammals, such as black rats, wild boars, and rabbits, carry *Leptospira* spp. on Amami Oshima Island. In particular, Ryukyu wild boars had a high

infection rate (58.3%) and were considered a maintenance host of *L. interrogans* serogroup Hebdomadis. Hunting of wild boars is popular on this island, and hunters often disjoint carcasses with their bare hands. In addition, injured Amami rabbits are commonly rescued and receive medical treatment at animal hospitals on the island. Moreover, the number of visitors has increased for river fishing, river kayaking, hiking in the forest, and other activities. Leisure in freshwater is a well-known risk factor for *Leptospira* infection [44,45]. Indeed, laboratory-confirmed leptospirosis patients in 2016 were either involved in leisure activities or working in a river [18,19]. Therefore, to prevent human leptospirosis on this island, rubber gloves should be worn when handling wild boars and rabbits. When entering the river, long-sleeved clothes should be worn to avoid wounding.

 *L. interrogans* serogroups Australis and Hebdomadis can cause lethal infections in dogs [13]; however, these serogroup strains are not included in canine vaccines for leptospirosis currently available in Japan. A significant increase in antibody titers against serogroup Hebdomadis in paired serum samples was observed in one dog on Amami Oshima Island in 2018 (S2 Table). Therefore, it is important to keep companion dogs away from environments potentially contaminated with *Leptospira* spp. to prevent canine leptospirosis.

## Supporting information

**S1 Fig. Maximum likelihood tree based on the concatenated sequences of seven housekeeping genes from *L. interrogans*.** The STs of serogroups Australis and Hebdomadis identified in Japan are highlighted in yellow and blue, respectively. The ST highlighted in green indicates that it is identified both in the serogroups Australis and Hebdomadis in Japan. The novel STs identified in this study are indicated in red font.
(TIF)

**S1 Table. Clinical characteristics and laboratory data of D-KS19-7K.**
(DOCX)

**S2 Table. Clinical characteristics and laboratory data of the female hunting dog confirmed as leptospirosis in Amami Oshima Island in 2018.**
(DOCX)

## Acknowledgments

We are grateful to the Amami Wildlife Center for capturing rats and Shinji Maruyama for handling the dogs and rats. We thank Yuki Masaki for providing the kidney tissues of wild boars and the Okinawa Prefectural Institute of Health and Environment for providing *Leptospira* DNA samples. We also thank Masatomo Morita for his technical assistance and Yojiro Shimada, Seira Uehara, Mayumi Ikeda, Megumi Suzuki, Kaho Sato, and Hinako Haga for their support throughout this study.

## Author Contributions

**Conceptualization:** Nobuo Koizumi.

**Data curation:** Nobuo Koizumi.

**Formal analysis:** So Shinya, Nobuo Koizumi.

**Funding acquisition:** Nobuo Koizumi.

**Investigation:** So Shinya, Daigo Negishi, Nobuo Koizumi.

**Methodology:** Nobuo Koizumi.

**Project administration:** So Shinya, Nobuo Koizumi.

**Resources:** So Shinya, Nobuo Koizumi.

**Software:** So Shinya, Nobuo Koizumi.

**Supervision:** Yukinori Muraoka.

**Validation:** So Shinya, Nobuo Koizumi.

**Visualization:** So Shinya, Nobuo Koizumi.

**Writing – original draft:** So Shinya, Nobuo Koizumi.

**Writing – review & editing:** So Shinya, Yukinori Muraoka, Daigo Negishi, Nobuo Koizumi.

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
