## [Decision Letter · Decision Letter 0]

5 Mar 2021

PONE-D-21-01288

Molecular epidemiology of Leptospira spp. among wild mammals and dogs in Amami Oshima Island, Japan

PLOS ONE

Dear Dr. Shinya,

Thank you for submitting your manuscript to PLOS ONE. After careful consideration, we feel that it has merit but does not fully meet PLOS ONE’s publication criteria as it currently stands. Therefore, we invite you to submit a revised version of the manuscript that addresses the points raised during the review process.

Please consider the comments of the reviewers to improve the manuscript.

We look forward to receiving your revised manuscript.

Kind regards,

Axel Cloeckaert

Academic Editor

PLOS ONE

Journal Requirements:

3. We note that Figure 1 in your submission contain map images which may be copyrighted. All PLOS content is published under the Creative Commons Attribution License (CC BY 4.0), which means that the manuscript, images, and Supporting Information files will be freely available online, and any third party is permitted to access, download, copy, distribute, and use these materials in any way, even commercially, with proper attribution. For these reasons, we cannot publish previously copyrighted maps or satellite images created using proprietary data, such as Google software (Google Maps, Street View, and Earth). For more information, see our copyright guidelines: http://journals.plos.org/plosone/s/licenses-and-copyright.

3.1.    You may seek permission from the original copyright holder of Figure 1 to publish the content specifically under the CC BY 4.0 license. 

3.2.    If you are unable to obtain permission from the original copyright holder to publish these figures under the CC BY 4.0 license or if the copyright holder’s requirements are incompatible with the CC BY 4.0 license, please either i) remove the figure or ii) supply a replacement figure that complies with the CC BY 4.0 license. Please check copyright information on all replacement figures and update the figure caption with source information. If applicable, please specify in the figure caption text when a figure is similar but not identical to the original image and is therefore for illustrative purposes only.

Reviewers' comments:

Reviewer's Responses to Questions

**Comments to the Author**

1. Is the manuscript technically sound, and do the data support the conclusions?

Reviewer #1: Partly

Reviewer #2: Yes

2. Has the statistical analysis been performed appropriately and rigorously? 

Reviewer #1: N/A

Reviewer #2: N/A

3. Have the authors made all data underlying the findings in their manuscript fully available?

Reviewer #1: No

Reviewer #2: Yes

4. Is the manuscript presented in an intelligible fashion and written in standard English?

Reviewer #1: Yes

Reviewer #2: Yes

5. Review Comments to the Author

Reviewer #1: Shinya et al. manuscript, “Molecular epidemiology of Leptospira spp. among wild mammals and dogs in Amami Oshima Island, Japan” describes for the first time the genetic variability of Leptospira spp. in a region with high risk of infection in Japan. The study is of local importance since there is an epidemiological context that benefits Leptospira spp. infection and nevertheless the population of this pathogen has not been previously studied.

Authors evaluated samples from 42 black rats (Rattus rattus), 12 Ryukyu wild boars (Sus scrofa riukiuanus), 2 goats (Capra aegagrus hircus) and 1 rabbit (Pentalagus furnessi) from different location within Amami Oshima Island.

Serogroup, species (single locus strategy) and intra-species variability (MLST or MLVA) were evaluated. Authors described that Leptospira spp. were isolated from 3 Ryukyu wild boars, 2 black rats, 1 rabbit and 1 dog.

Intra-species variability was evaluated by 2 different strategies according to the species. L. borgpetersenii was evaluated by MLST #1 and L. interrogans by MLVA as previous studies of this group.

Data accession: only flaB sequences were deposited in DDBJ, MLST and MLVA sequences should also be available.

General suggestions by the reviewer:

Previous studies by the authors included MLVA typing of L. borgpetersenii and L. interrogans isolates from different regions of Japan (https://doi.org/10.1016/j.meegid.2015.08.013). On the other hand, pubmlst database includes 30 isolates from Japan with information about STs variability according to the same scheme applied in this work (https://pubmlst.org/bigsdb?db=pubmlst_leptospira_isolates&page=query, search: country=Japan).

Among these isolates the database show 15 STs with 4 STs assigned to L. borgpetersenii and 10 to L. interrogans. In this context, my suggestion is to include MLST typing to L. interrogans isolates to better understand the population genetics proposed in the study as MLST strategy is based on loci under negative selection.

Furthermore, MLST typing should be included in the pubmlst database.

On the other hand, MLVA is more accurate to study a specific location in a specific period of time since these markers variate more frequently. Therefore, these results is adequate for the Amami Oshima Island as a picture of genetic variability during the study.

Overall, the study is missing a phylogenetic study in order to better understand the relations among the new isolates as commented in the discussion section. Please add this analysis in the context of previous data.

Specific changes suggested by the reviewer:

Page 7, line 139: the first reference in the line should be 19, not 20.

Page 8, line 165: delete “(DDBJ 165 accession number LC596935– LC596941)” as this is already mentioned in methods section.

Page 12, line 210: replace “..black rats are an important reservoir of L. interrogans for dogs on Amami Oshima Island” to “…black rats may be an important reservoir of L. interrogans for dogs on Amami Oshima Island”. The number of isolates studied is too small to make a statement.

Page 13, line 213: authors cannot speak of prevalence because the study does not include a sampling design with a distribution analysis. Change prevalence to frequency

Page 13, line 216: change various to different

Table 2 and supplementary table 1: both tables should be merged. Supplementary table specifies as reference “this study” for isolates not described in the manuscript, please correct this.

Reviewer #2: This a well written and straight forward manuscript providing valuable epidemiological information on Leptospira prevalence supported by isolation and molecular characterization, in Amami Oshima island in Japan.

Authors must add approval information for animal use protocol used in this study

Line 83 What do you mean by breeding places?

6. PLOS authors have the option to publish the peer review history of their article (what does this mean?). If published, this will include your full peer review and any attached files.

Reviewer #1: **Yes: **Paula Ruybal

Reviewer #2: No

---

## [Author Response · Author response to Decision Letter 0]

26 Mar 2021

Response to Editorial Office and Reviewers

Comments from Editorial Office

Comment 1: Please ensure that your manuscript meets PLOS ONE's style requirements, including those for file naming. 

Reply: We revised our manuscript according to the PLOS ONE's style requirements. We also had our revised manuscript checked by professional proofreading.

Comment 2: We note that you have included the phrase “data not shown” in your manuscript. Unfortunately, this does not meet our data sharing requirements. PLOS does not permit references to inaccessible data. We require that authors provide all relevant data within the paper, Supporting Information files, or in an acceptable, public repository. Please add a citation to support this phrase or upload the data that corresponds with these findings to a stable repository (such as Figshare or Dryad) and provide and URLs, DOIs, or accession numbers that may be used to access these data. Or, if the data are not a core part of the research being presented in your study, we ask that you remove the phrase that refers to these data.

Reply: We prepared a Supporting Information file (S1 Table) for the laboratory data on the dog with leptospirosis in Amami Oshima Island in 2018 that was described as “data not shown” in our previous manuscript. We also prepared a Supporting Information file (S2 Table) for the laboratory data on D-KS19-7K.

We also cited the URL for the report on leptospirosis patients in Japan in 2016 in our revised manuscript for the “unpublished data” in our previous manuscript.

Comment 3: We note that Figure 1 in your submission contain map images which may be copyrighted. All PLOS content is published under the Creative Commons Attribution License (CC BY 4.0), which means that the manuscript, images, and Supporting Information files will be freely available online, and any third party is permitted to access, download, copy, distribute, and use these materials in any way, even commercially, with proper attribution. For these reasons, we cannot publish previously copyrighted maps or satellite images created using proprietary data, such as Google software (Google Maps, Street View, and Earth). 

Reply: We prepared Figure 1 using pictures published by the Geospatial Information Authority of Japan that are provided under the CC BY 4.0.

Comments from Reviewer #1

Comment 1: only flaB sequences were deposited in DDBJ, MLST and MLVA sequences should also be available.

Reply: We deposited the isolate information and MLST profiles in the PubMLST database. We added the following sentence in P7L141. 

Information on the isolates analyzed in this study and new STs (ST328, ST329, and ST331) obtained in this study have been deposited in the database.

We also deposited genome sequences containing VNTR determined in this study in DDBJ (accession numbers LC619028–LC619057). We added the following sentence in P7L154. 

The genome sequences containing VNTR determined in this study have been deposited in a public database (DDBJ accession numbers LC619028–LC619057).

The MLVA profiles for the Leptospira strains isolated in Okinawa Prefecture had previously been determined by the comparison of those from Leptospira strains in our previous study [Ref 14: Koizumi et al., Infect Gen Evol. 2015].

Comment 2: Previous studies by the authors included MLVA typing of L. borgpetersenii and L. interrogans isolates from different regions of Japan (https://doi.org/10.1016/j.meegid.2015.08.013). On the other hand, pubmlst database includes 30 isolates from Japan with information about STs variability according to the same scheme applied in this work (https://pubmlst.org/bigsdb?db=pubmlst_leptospira_isolates&page=query, search: country=Japan).

Among these isolates the database show 15 STs with 4 STs assigned to L. borgpetersenii and 10 to L. interrogans. In this context, my suggestion is to include MLST typing to L. interrogans isolates to better understand the population genetics proposed in the study as MLST strategy is based on loci under negative selection.

Furthermore, MLST typing should be included in the pubmlst database.

On the other hand, MLVA is more accurate to study a specific location in a specific period of time since these markers variate more frequently. Therefore, these results is adequate for the Amami Oshima Island as a picture of genetic variability during the study.

Overall, the study is missing a phylogenetic study in order to better understand the relations among the new isolates as commented in the discussion section. Please add this analysis in the context of previous data.

Reply: We performed MLST for L. interrogans serogroups Australis and Hebdomadis strains isolated in Kyushu Main Island, Okinawa Prefecture, and Taiwan. STs determined were included in Table 2 in our revised manuscript. We also prepared the phylogenetic tree based on the concatenated sequences of the seven genes as a supplementary figure (S1 Fig). For these, we modified the paragraph of MLST in the Materials and Methods section as follows:

(Previous) MLST was performed for the Leptospira borgpetersenii isolate using seven housekeeping genes, glmU, pntA, sucA, tpiA, pfkB, mreA, and caiB, as previously described [20]. Sequence type (ST) was assigned through the PubMLST database [20]. 

(Revised) MLST was performed for the Leptospira strains isolated in this study, for those isolated in Kyushu Main Island, and those isolated in Okinawa Prefecture (Table 2) using seven housekeeping genes, glmU, pntA, sucA, tpiA, pfkB, mreA, and caiB, as previously described [20]. Sequence type (ST) was assigned using the PubMLST database [21]. Information on the isolates analyzed in this study and new STs (ST328, ST329, and ST331) obtained in this study have been deposited in the database. Phylogenetic relationships of the concatenated sequences (5ʹ-glmU-pntA-sucA-tpiA-pfkB-mreA-caiB-3ʹ) determined in this study and deposited in the MLST database (PubMLST, https://pubmlst.org/organisms/leptospira-spp) were determined by reconstructing a phylogenetic tree by the maximum likelihood method using IQ-TREE [22] with 1,000 ultrafast bootstrap replicates.

We described the results of MLST in the Results section as follows:

P10L181. The rat and dog isolates of L. interrogans serogroup Australis (AMM-012 and D-KS19-7K) belonged to ST105, which was detected in Okinawa Prefecture and Taiwan, whereas the L. interrogans serogroup Australis strain isolated from Kyushu Main Island belonged to the phylogenetically distant lineage, ST120 (Table 2 and S1 Fig). Two STs were identified in L. interrogans serogroup Hebdomadis isolates (Table 2). Two of the three Ryukyu wild boar isolates (AMM-042 and AMM-043) showed the novel ST, ST328, which was phylogenetically related to other STs of serogroup Hebdomadis strains in Japan (S1 Fig). The other wild boar isolate (AMM-047) and the Amami rabbit isolate (AMM-057) belonged to ST140, which has been identified in L. interrogans serogroup Hebdomadis strains on Kyushu Main Island (Table 2).

We also modified the descriptions of MLVA results as follows:

(Previous) MLVA was employed for L. interrogans because MLVA using 11 loci has a higher discriminatory power and was more concordant with serotyping than MLST using seven housekeeping genes for L. interrogans [13, 14]. As a result, the rat and dog isolates of L. interrogans serogroup Australis (AMM-012 and KS19-7K) showed the same MLVA profile (MLVA type) (Table 2). The profile was almost identical to that of the isolates from Okinawa Prefecture and Taiwan (identity: 10/11, S1 Table), whereas it was different from those from Kyushu Main Island (identity: ~1/11) [13, 14]. Two MLVA types were identified in L. interrogans serogroup Hebdomadis isolates from Ryukyu wild boars, one of which (AMM-047) was identical to that from the Amami rabbit (Table 2). One of the MLVA types (AMM-047 and AMM-057) was identical to that of the isolates from Kyushu Main Island, whereas it was different from those from Okinawa Prefecture (identity: ~3/11, S1 Table) [13, 14]. Another MLVA type (AMM-042 and AMM-043) was different from those of the isolates from Kyushu Main Island and Okinawa Prefecture (identity: ~4/11, S1 Table) [13, 14].

 (Revised) MLVA was further employed for L. interrogans isolates because MLVA using 11 loci has higher discriminatory power and was more concordant with serotyping than MLST using seven housekeeping genes for L. interrogans [13,14]. The rat and dog isolates of L. interrogans serogroup Australis (AMM-012 and D-KS19-7K) showed the same MLVA profile (MLVA type) (Table 2). The profile was almost identical to that of the isolates from Okinawa Prefecture and Taiwan (identity: 10/11, Table 2), whereas it was different from those from Kyushu Main Island (identity: ~1/11) (Table 2) [13,14]. As with MLST, two MLVA types were identified in L. interrogans serogroup Hebdomadis isolates: the wild boars and rabbit isolates (AMM-047 and AMM-057) showed identical MLVA profiles, which were identical to those of the isolates from Kyushu Main Island, whereas they were different from those from Okinawa Prefecture (identity: ~3/11, Table 2) [13,14]. The MLVA types of AMM-042 and AMM-043 were different from those of the isolates from Kyushu Main Island and Okinawa Prefecture (identity: ~4/11, Table 2) [13,14].

For the interpretation of MLST results, we added the following sentences in the Discussion section.

P19L303. On the other hand, the same ST, ST118, was identified in human and dog patients both in Kyushu Main Island and Okinawa Prefecture (Table 2), suggesting that there is an unidentified maintenance animal(s) that carries this ST in these areas. This may also be true for the MLVA type/ST140 of the serogroup Hebdomadis strain. In addition, novel ST/MLVA profiles were identified in wild boar isolates of the serogroup Hebdomadis. Further research is needed to clarify the phylogenetic relationship of L. interrogans serogroup Hebdomadis strains in Japan.

We included the results of MLST in the Abstract of our revised manuscript too.

(Revised) Leptospirosis is a worldwide zoonosis caused by the pathogenic Leptospira spp. Canine and human leptospirosis sometimes occur on Amami Oshima Island, located in the Nansei Archipelago, southwestern Japan; however, information on the causative Leptospira spp. on this island is quite limited. This study aimed to investigate the molecular and serological characteristics of Leptospira spp. isolated from wild animals and a dog in Amami Oshima Island. 

We obtained seven Leptospira strains by culturing kidney tissues of wild animals, such as black rats (2), wild boars (3), and rabbit (1) as well as blood from a symptomatic dog. Using flaB sequencing and microscopic agglutination test with antisera for 18 serovars, the isolates were identified as Leptospira borgpetersenii serogroups Javanica (black rat), L. interrogans serogroup Australis (black rat and dog), and L. interrogans serogroup Hebdomadis (wild boar and rabbit). The sequence type (ST) of L. borgpetersenii serogroup Javanica was determined to be ST143 via multilocus sequence typing (MLST) using seven housekeeping genes. For L. interrogans, MLST and multiple-locus variable-tandem repeat analysis (MLVA) revealed identical ST and MLVA types in rat and canine isolates, whereas two STs and MLVA types were identified in wild boar isolates. The STs and MLVA types of rabbit and one of the wild boars were identical. Bacterial culture and flaB-nested polymerase chain reaction demonstrated a high rate of Leptospira infection in wild boars (58.3%, 7/12), whereas Leptospira spp. were detected in 4.8% of black rats (2/42). 

This study revealed diverse Leptospira genotype and serotype maintenance in wild mammals on Amami Oshima Island. MLST and MLVA indicated that black rats were a source of canine infection. Wild boars carry L. interrogans and are considered an important maintenance host because antibodies against serogroup Hebdomadis were detected in human and canine leptospirosis patients on this island.

Comment 3: Page 7, line 139: the first reference in the line should be 19, not 20.

Reply: Thank you for the reviewer’s indication. We corrected the reference number. 

Comment 4: Page 8, line 165: delete “(DDBJ 165 accession number LC596935– LC596941)” as this is already mentioned in methods section.

Reply: According to the reviewer’s comment, we deleted the phrase in our revised manuscript. 

Comment 5: Page 12, line 210: replace “..black rats are an important reservoir of L. interrogans for dogs on Amami Oshima Island” to “…black rats may be an important reservoir of L. interrogans for dogs on Amami Oshima Island”. The number of isolates studied is too small to make a statement.

Reply: We changed the sentence as the reviewer suggested.

Comment 6: Page 13, line 213: authors cannot speak of prevalence because the study does not include a sampling design with a distribution analysis. Change prevalence to frequency.

Reply: According to the reviewer’s comment, we changed the word “prevalence” to “frequency”.

Comment 7: Page 13, line 216: change various to different

Reply: According to the reviewer’s comment, we changed the word “various” to “different”.

Comment 8: Table 2 and supplementary table 1: both tables should be merged. Supplementary table specifies as reference “this study” for isolates not described in the manuscript, please correct this.

Reply: According to the reviewer’s comment, we prepared the new Table 2 by merging the previous Table 2 and S1 Table. As mentioned above, we added data on L. interrogans serogroups Australis and Hebdomadis strains isolated in Kyushu Main Island in the revised Table 2. 

Comments from Reviewer #2 

Comment 1: Authors must add approval information for animal use protocol used in this study

Reply: The rats were captured as the program of the Ministry of the Environment and this investigation was approved by the Ministry. We modified the sentence as follows: 

(Previous) A total of 42 black rats (Rattus rattus) were captured as a bycatch under the extermination program of invasive feral cats conducted by the Ministry of the Environment.

(Revised) A total of 42 black rats (Rattus rattus) were captured as a bycatch under the extermination program of invasive feral cats conducted by the Ministry of the Environment, and this investigation was approved by the Ministry.

Cultivation of Leptospira spp. from dead Amami rabbits was also approved by the Ministry of the Environment. We added the following sentence in our revised manuscript.

P5L92. The cultivation of Leptospira spp. from dead Amami rabbits was also approved by the Ministry.

In Japan, there is no ethics committee available for private-practice animal hospitals. The investigation on canine leptospirosis was conducted according to the ethical codes of the Japan Veterinary Medical Association. We obtained the oral consent for collection of blood for Leptospira cultivation from owners (P5L99). 

Comment 2: Line 83 What do you mean by breeding places?

Reply: We used “breeding place” as the places where the dogs enrolled in this study were kept by their owners. We change the word “breeding place” into “place of residence” in the revised manuscript.

---

## [Editor Report · Decision Letter 1]

30 Mar 2021

Molecular epidemiology of Leptospira spp. among wild mammals and dogs in Amami Oshima Island, Japan

PONE-D-21-01288R1

Dear Dr. Shinya,

We’re pleased to inform you that your manuscript has been judged scientifically suitable for publication and will be formally accepted for publication once it meets all outstanding technical requirements.

Kind regards,

Axel Cloeckaert

Academic Editor

PLOS ONE
---

## [Editor Report · Acceptance letter]

1 Apr 2021

PONE-D-21-01288R1 

Molecular epidemiology of *Leptospira* spp. among wild mammals and a dog in Amami Oshima Island, Japan 

Dear Dr. Shinya:

I'm pleased to inform you that your manuscript has been deemed suitable for publication in PLOS ONE. Congratulations! Your manuscript is now with our production department. 

Kind regards, 

on behalf of

Dr. Axel Cloeckaert 

Academic Editor

PLOS ONE